# In Silico Multi-Target Approach Revealed Potential Lead Compounds as Scaffold for the Synthesis of Chemical Analogues Targeting SARS-CoV-2

**DOI:** 10.3390/biology11030465

**Published:** 2022-03-18

**Authors:** Alfonso Trezza, Claudia Mugnaini, Federico Corelli, Annalisa Santucci, Ottavia Spiga

**Affiliations:** Department of Biotechnology, Chemistry and Pharmacy (Department of Excellence 2018–2022), University of Siena, 53100 Siena, Italy; federico.corelli@unisi.it (F.C.); annalisa.santucci@unisi.it (A.S.); ottavia.spiga@unisi.it (O.S.)

**Keywords:** docking simulation, SARS-CoV-2 spike protein, 3CLpro, COVID-19, multi-target inhibitors, homology modeling, molecular dynamics simulation

## Abstract

**Simple Summary:**

COVID-19 is an infectious disease caused by SARS-CoV-2. The virus has rapidly spread to humans, causing the ongoing coronavirus pandemic. Enormous progress in finding therapies has been made, but an effective therapy is still absent. In this study, we propose a computational strategy aimed at identifying novel multi-target scaffolds against the virus. The proposed study was performed using bioinformatics methods, such as homology modeling, virtual screening and classical molecular dynamics, where 3D structures of targets were reconstructed and compounds with potential inhibitory activity against different virus targets were identified. Furthermore, a potential mechanism of action was proposed. Our study could provide new insights and approaches for the rapid identification of novel multi-target inhibitors of SARS-CoV-2.

**Abstract:**

Severe acute respiratory syndrome-coronavirus 2 (SARS-CoV-2) causes coronavirus disease 2019 (COVID-19), an infectious disease that spreads rapidly in humans. In March 2020, the World Health Organization (WHO) declared a COVID-19 pandemic. Identifying a multi-target-directed ligand approach would open up new opportunities for drug discovery to combat COVID-19. The aim of this work was to perform a virtual screening of an exclusive chemical library of about 1700 molecules containing both pharmacologically active compounds and synthetic intermediates to propose potential protein inhibitors for use against SARS-CoV-2. In silico analysis showed that our compounds triggered an interaction network with key residues of the SARS-CoV-2 spike protein (S-protein), blocking trimer formation and interaction with the human receptor hACE2, as well as with the main 3C-like protease (3CLpro), inhibiting their biological function. Our data may represent a step forward in the search for potential new chemotherapeutic agents for the treatment of COVID-19.

## 1. Introduction

The coronavirus disease 2019 (COVID-19), caused by the severe acute respiratory syndrome coronavirus 2 (SARS-CoV-2), is an infectious disease that has spread rapidly all over the world since the last months of 2019. The World Health Organization (WHO) declared a COVID-19 pandemic on 12 March 2020 due to the levels of spread and severity. The process of SARS-CoV-2 entering into the host cell starts with the attachment of the viral S-glycoprotein trimer to host cell angiotensin-converting enzyme 2 receptor (ACE2) [1]. This binding process allows: (I) the fusion of the viral membrane with the host cell; (II) the viral entry; (III) the release of its genomic material in the cytoplasm to be translated in the cell [2]. The viral mRNA contains two overlapping open reading frames, encoding the pp1a and pp1ab polyproteins [3], which are subsequently cleaved by 3CLpro, producing non-structural proteins (nsps) 1–11 and 1–16, respectively [4]. Nsps are crucial for several viral processes, such as the life cycle and the virulence of the virus [5]. The infection is transmitted via the inhalation of contaminated droplets, but also by contact of contaminated hands with the nasal mucosa or with the eyes [6]. Coronaviruses hit the respiratory tract via the nose. The most common symptoms, which usually occur after three days of incubation, are a cold, runny nose, cough, loss of smell and nasal obstruction [7]. The disease ends in a few days, but some types of patients, especially the elderly and people with co-existing illness [8], show shortness of breath, which in some cases causes death [9]. The COVID-19 outbreak is having a dramatic socio-economic impact. People in many countries stopped working and travelling for a short period, and people were asked to change their habits and behavior because of the restrictive social distancing measures. The world-wide approach against the spread of the virus was quarantining people. Currently, gatherings are not allowed yet, and life has changed. The crisis has not finished yet, and this is the reason why researchers from all over the world are progressively trying to find a conclusive cure to stop the virus. Several repurposed antiviral drugs are under investigation or in the early stages of regulatory approval for the treatment of COVID-19 [10].

No other approved drug-like molecules specifically targeting the virus have been released, and the global approach to drug discovery, aimed at decreasing the time and costs of this process, is primarily focused on repurposing FDA-approved drugs. Drug repurposing is a modern drug development strategy used to detect novel uses for drugs approved by the FDA outside the range of their original medical suggestion [11]. It aims at establishing whether an ‘old drug’ can be reused for innovative therapeutic purposes, representing an quicker and more economical alternative to de novo drug discovery development, which generally takes 12–15 years and costs USD 2–3 billion (from production to approval, passing through the various phases of preclinical and clinical trials) [11]. Huge progress and significant results have been obtained to identify and propose therapeutic agents for the treatment of COVID-19. Unfortunately, no conclusive cure is available for the infection by SARS-CoV-2. Presently, there are several encouraging therapeutic candidates, that are being evaluated against COVID-19 with emergency-use authorization by the FDA and NIH, such as remdesivir, an FDA-approved inhibitor [10] of viral RNA synthesis that acts by inhibiting the RNA-dependent RNA polymerases (RdRps) and competes with adenosine triphosphate [12]; lopinavir–ritonavir, protease enzyme inhibitors that bind to Mpro, a key enzyme for coronavirus replication [13]; and baricitinib and ruxolitinib, inhibitors of Janus kinases enzymes that alleviate the signal transmission due to the cytokine storm [14].

In fact, according to recent preliminary results of randomized clinical trials, these compounds have been shown to be superior to placebos in shortening the time to recovery in adults hospitalized with COVID-19 and with evidence of lower respiratory tract infection [14]. Nevertheless, the mortality remains high, showing that treatment with an antiviral drug alone is not likely to be enough [15,16,17]. To continue to improve patient outcomes in COVID-19, combinations of diverse antiviral agents or of antiviral agents with drugs that have other clinical indications should be evaluated for future strategies.

In this work, the in silico analysis of an exclusive property chemical library of around 1700 molecules, including both pharmacologically active compounds and synthetic intermediates, has been performed. A few structural clusters, such as 4-quinolones, 4-hydroxy-2-quinolones and indoles, can be identified in the library, which represent privileged medicinal chemistry scaffolds [18]. Interestingly, a large number of molecules within this library belong to the family of cannabinoid receptor 2 ligands, which have recently been proposed as a therapeutic option for SARS-CoV-2 infections for their ability to limit the release of pro-inflammatory cytokines, shift the macrophage phenotype towards the anti-inflammatory M2 type and enhance the immune-modulating properties of mesenchymal stromal cells [19]. This aspect is particularly relevant because it could, in principle, allow us to combine virus inhibition at multiple targets with the treatment of the effects of virus infection that are mainly inflammatory and coagulative. Our study identified novel potential quinolone- and indole-based inhibitors able to simultaneously block viral S-protein trimerization and the interaction of SARS-CoV-2 with the S-protein/hACE2. Furthermore, other compounds of our library inhibit the viral 3CLpro. In this work, we propose potential lead compounds as scaffolds for the synthesis of chemical analogues targeting SARS-CoV-2.

## 2. Materials and Methods

### 2.1. Chemistry

The 4-quinolones COR480, COR482 and COR483 had previously been synthesized and evaluated as cannabinoid receptor ligands [20]. Similarly, the indole derivative COR437 had been already described by us [21]. Conversely, compounds COR267, COR1393, and COR1461 had never been reported before and their synthesis is described in the Appendix A.

### 2.2. In Silico Methods

#### 2.2.1. Optimization of Model 3D Structures 

The Cryo-EM structure of the 2019-nCoV spike in the prefusion conformation was downloaded from the RCSB Protein Data Bank [22] (PDB code 6VSB) [23] with UniProtKB P0DTC2. The model was devoid of N-acetyl glucosamine (NAG) glycan residues and consisted of the glycoprotein trimer form where each monomer had amino acids ranging from 27 to 1146. The structure of the novel coronavirus spike receptor-binding domain complexed with its receptor ACE2 was obtained from RCSB Protein Data Bank with PDB code 6LZG [24] and UniProtKB Q9BYF1. From the model, we removed the N-acetylglucosamine residues, which are not located in the RBD pocket [25]. SARS-CoV-2 3CL protease (3CL pro) in a complex with a novel inhibitor was downloaded from the RSCB Protein Data Bank [22] with PDB code 6M2N [26], and its primary structure was retrieved from the Uniprot database [27] with UniProtKB P0DTD1. Co-crystallographic ligands and water molecules were all removed from the original PDB files and non-polar hydrogens were merged. To resolve errors during the molecular docking simulation, possible missing side chains and steric clashes in the 6M2N file were added and optimized through homology modeling, carried out using PyMOD 2.0 [28], with CLUSTALO as the sequence alignment tool and Modeller as the homology modeling tool, both implemented in PyMOD2.0. Then, the 3D structure was validated using PROCHECK [29].

#### 2.2.2. Energy Minimization and the Relaxation of Model 3D Structures

GROMACS 2019.3 [30] with charmm36-mar2019 force field [31] was used to minimize unfavorable intramolecular interactions before running the docking simulation by applying 10.000 steps of minimization with the steepest descent algorithm, converging to a minimum energy with forces less than 100 kJ/mol/nm. The charge of each residue was assigned as default by the charmm36-mar2019 force field. In order for the protein to avoid seeing its image across the periodic boundary, the structure was immersed in a cubic box with a dimension twice that of the cutoff distance from the next nearest image of itself. Then, the TIP3P water model was chosen to fill the box. Counter ions were added to the systems to balance the protein net charge. The simulation was performed in periodic boundary conditions. The energy of the system was minimized as mentioned above. The LINCS algorithm constrained the bond lengths involving hydrogen atoms. To keep the temperature constant at 310 K and the pressure at 1 atm, we performed 1ns of NVT ensemble using a V-rescale thermostat (with a time constant of 0.1 ps) and more cycles (using a python script written in house) of isothermal–isobaric ensemble lasting 5 ns with a Berendsen barostat (with a time constant of 0.1 ps). The short-range electrostatic and van der Waals cutoff were set to 1.0 nm. Each conformation was collected every 10 ps, setting a time step of 0.002 ps. To relax the biological system, it was subject to a short 10 ns classical molecular dynamics (cMD) simulation. 

#### 2.2.3. Docking Simulation

A virtual screening using a library of 1764 compounds, synthetized in house, was carried out using AutoDock/VinaXB [32]. A box was created around the binding pocket of 6VSB and 6LZG, involving all binding residues proposed by Bongini P. et al. [33] and Trezza A. et al. [34], respectively, while for the 3CLpro virtual screening, a box with dimensions of 30, 30 and 30 Å was created around the binding pocket of a known inhibitor of 6M2N [26]. From the virtual screening process, we achieved a large array of possible ligands; hence, to restrict the binding poses obtained with the docking simulation and to use a standard for each simulation, we considered only the best three candidates on the basis of their binding free energy on the target. The best compounds were selected for further in silico investigations. MGLTOOLS scripts [35] and OpenBabel [36] were used to convert the protein and ligand files (the ligands library was drawn with the ChemDraw tool and saved in sdf format) and to add Gasteiger as partial charges and all missing hydrogen atoms. LigPlot++ v.2.2.4 [37] analyzed the target–ligand interaction network. 

#### 2.2.4. Classical Molecular Dynamics (cMD) Simulation

A 100 ns classical molecular dynamics (cMD) simulation was used for each biological system, as mentioned above. The strength of the interaction between targets and ligands was evaluated according to the non-bonded interaction energy, using the gmx energy package in GROMACS 2019.3, selecting the short-range Coulombic interaction energy (Coul-SR) and the short-range Lennard-Jones energy (LJ-SR) as energy terms of interest. The total interaction energy (IE_Binding_) is defined by:IE_Binding_ = Coul-SR + LJ-SR(1)

All cMD analyses were performed with GROMACS 2019.3 package. Opensource PyMOL v.2.2 was used for structural visualization and creating the images.

## 3. Results

In the following subheadings, we describe a structure-based virtual screening of SARS-CoV-2 S-glycoprotein and viral 3CLpro with a database of 1764 in-house-produced compounds to identify potential SARS-CoV-2 inhibitors. A docking simulation of the S-glycoprotein was carried out on two different binding regions, namely: (I) the region involved in the trimerization process, to prevent its biological assembly; and (II) the SARS-CoV-2 S-glycoprotein RBD/hACE2 binding region, to avoid their interaction and subsequent viral/human membrane fusion. Finally, a virtual screening was conducted on the 3CLpro to block protease biological function. To restrict the docking results, we selected only the best three compounds for each target, on the basis of their binding free energy (the 2D molecular structures are reported in Appendix A). However, to show the reliability of hit compound selection, we provided the distribution of top 100 binding free energy scores (Appendix A) of ligands within the binding pocket of the target. The interaction network of each ligand on the target was evaluated to elucidate the binding mode. A cMD simulation was performed for each biological system (the snapshots of models used for each of the simulations are reported in Appendix A) in order to confirm the docked pose stability of ligands and their interaction energy with the target after 100 ns run. The computational workflow applied in this study is reported in Appendix A.

### 3.1. SARS-CoV-2 S-Glycoprotein Trimerization Region: Virtual Screening and cMD

To define the potential binding pose of ligands on the S-glycoprotein, we performed a docking simulation in the binding region proposed by Bongini P. et al. [19]. In the virtual screening results, the compounds COR480, COR482 and COR483 exhibited the lowest binding free energy on the target (−8.8 Kcal/mol, −8.8 Kcal/mol and −8.3 Kcal/mol, respectively), sharing the same binding pocket and binding pose (Figure 1A). 

Interaction network analyses of each ligand within the target binding pocket showed that COR480 formed two halogen bonds involving its fluorine and chlorine atoms with Ser-884 Oy and Phe-898, respectively, and gave rise to six hydrophobic interactions with Tyr-789, Lys-790, Phe-797, Gly-880, Thr-883 and Ala-893 (Figure 1B). COR482 was involved in the same halogen bonds as COR480 and six hydrophobic interactions with Tyr-789, Lys-790, Pro-792, Phe-797, Gly-880, Thr-883 and Ala-893 (Figure 1C). Finally, COR 483 triggered one halogen bond between its chlorine atom and the nitrogen of Phe-898, as well as nine hydrophobic interactions with Gln-787, Tyr-789, Lys-790, Pro-792, Phe-797, Ala-879, Gly-880, Thr-883 and Ala-893 (Figure 1D). To confirm binding pose stability, 100 ns cMD simulations were run for the SARS-CoV-2 S-glycoprotein trimer alone and in complexes with the compounds. To evaluate the validity of the cMD protocol, we examined the protein structural integrity during the simulations. The small differences between the backbone RMSD of the apo protein (12 Å) and the modeling of a ligand in bound form (from 10 Å to 15 Å) did not add significant artefacts in the cMD simulations (Appendix A). The RMSD average of the ligands was between 1.5 Å and 2.5 Å (Appendix A), showing good docked pose stability. To quantify the strength of the interaction between ligands within the trimer binding site, the interaction energy between the protein and the ligands was evaluated. The total interaction energy for COR480, COR482 and COR483 exhibited −137.27 ± 12.8 kJ/mol, −250.8 ± 6.9 kJ/mol and −189.5 ± 5.6 kJ/mol, respectively. These data suggest that ligands bind spontaneously to the target with high affinity, confirming the docking study.

### 3.2. SARS-CoV-2 RBD/hACE2 Interaction Region: Virtual Screening and Cmd

The virtual screening of our library was conducted in the same viral spike-RBD binding pocket proposed by Trezza A. et al. [34]. The ligands COR480, COR482 and COR1393 showed the lowest binding free energy scores (−8.7 Kcal/mol, −8.6 Kcal/mol and −8.5 Kcal/mol, respectively).

The three ligands docked in the same binding pocket (Figure 2A) formed the following interaction network: COR480 was involved in a halogen bond between the oxygen atom (O1) of the carbonyl group and the hydroxyl of the Tyr-453 side chain, and the oxygen atom (O2) of the other carbonyl group and the Arg-403 Nϵ. Moreover, COR482 also had four hydrophobic interactions with Lys-417, Tyr-495, Gly-496, Phe-497 and Tyr-505 (Figure 2C). COR1393 triggered a hydrogen bond between its pyridine nitrogen and the guanidine of the Arg-403 side chain, in addition to six hydrophobic interactions with Lys-417, Tyr-453, Gln-493, Gly-496, Phe-497 and Gln-498 (Figure 2D). Small differences were observed between the backbone RMSD of the apo protein (1.8 Å) and the modeling of a ligand in bound form (from 1.5 Å to 4 Å), suggesting the presence of no added significant artefacts in the cMD simulations (Appendix A). To evaluate binding pose stability, 100 ns cMD simulations were performed for compounds in complexes with the SARS-CoV-2 RBD. The RMSD average of ligands was between 2 Å and 3 Å (Appendix A), confirming the stability of the binding pose. To quantify the strength of the interactions established by the ligands inside the target binding pocket, the interaction energy between the protein and the compounds was evaluated. The total interaction energy for COR480, COR482 and COR1393 was −178 ± 15.5 kJ/mol, −158.7 ± 9.1 kJ/mol and −108.2 ± 4.4 kJ/mol, respectively. These data suggest these ligands bind spontaneously to the target with high affinity, as shown by docking study. 

### 3.3. CLpro: Virtual Screening and cMD

To add strength and reliability to our results, we chose two different approaches to closely examine the binding of our compounds to the target. We analyzed the docking and molecular dynamics profiles of our compounds using both a target-based approach, by comparing them to a known 3CLPro inhibitor (baicalein), and an evolutionary approach, by selecting lopinavir and ritonavir, compounds with strong anti-SARS-CoV and MERS activity [38,39,40]. The sequence alignment of SARS-CoV-2 3CLpro shows that the SARS-CoV-2 proteinase is highly conserved compared to that of SARS-CoV-1, with an extremely high sequence identity of 96.1% [41]; moreover, 3CLpro shares a similar common cleavage site among coronaviruses [42]. COR267, COR437 and COR1461 were found to be the ligands with the best values of binding free energy (−9.4 Kcal/mol, −8.8 Kcal/mol and −8.8 Kcal/mol, respectively). Interestingly, lopinavir and ritonavir exhibited a lower binding free energy than our compounds (−8.0 Kcal/mol and −6.4 Kcal/mol, respectively).

Despite their structural and physicochemical diversity, the compounds bound in a similar mode (Figure 3A) shared the same binding pocket of lopinavir and ritonavir (Appendix A). In detail, COR267 was involved in a hydrogen bond between the hydrogen of pyrrole nitrogen and the carbonyl oxygen of Phe-140, as well as in eight hydrophobic interactions with His-41, Met-49, Leu-141, Met-165, Glu-166, Asp-187, Arg-188 and Gln-189 (Figure 3B). COR437 was implicated in a hydrogen bond between the oxygen of the amide group and the hydrogen of Asn-142 Nδ2 and in twelve hydrophobic interactions with His-41, Ser-46, Cys-44, Thr-45, Met-49, Pro-52, Leu-141, Cys-145, Met-165, Glu-166, Arg-188 and Gln-189 (Figure 3C). COR1461 triggered a strong hydrophobic network with Thr-25, Leu-27, His-41, Met-49, Phe-140, Leu-141, Cys-145, Met-165, Glu-166 and Gln-189 (Figure 3D). To confirm a reliable cMD simulation, we computed the backbone structural integrity during the simulations. The RMSD profiles of the apo protein (1.6 Å) and the ligand-bound target (from 1 Å to 3) were comparable, which demonstrates the good backbone structural stability of the systems run in the cMD simulations (Appendix A). To estimate the docked pose stability, 100 ns cMD simulations were performed for 3CLpro in complexes with the compounds. The RMSD average of systems of ligands was between 1 Å and 3 Å (Appendix A), indicating the reliability of the docked pose. Furthermore, the interaction energy between the protein and the ligands was also evaluated, resulting in values of −118.1 ± 22.6 kJ/mol, −167.5 ± 6.3 kJ/mol and −242 ± 16.6 kJ/mol for COR267, COR437 and COR1461, respectively. To confirm further the potential inhibitory activity of our compounds, we compared them to lopinavir and ritonavir dynamics profiles. From the MD simulation results, we noted that our compounds showed structural stability both in the backbone and in the ligand (Appendix A) as well as a similar interaction energy compared to lopinavir and ritonavir (−233.72 ± 3.14 kJ/mol and −166 ± 13.5 kJ/mol, respectively). These data suggest that ligands bind spontaneously to the target with high affinity, according to the docking study. 

## 4. Discussion

We performed a virtual in silico screening of an exclusive chemical library of about 1700 molecules against viral S-glycoprotein and 3CLpro to identify potential lead compounds that can act as scaffolds for the synthesis of chemical analogues against SARS-CoV-2. From the results of the virtual screening, we selected the top 100 compounds based on their free energy of binding to the target (Appendix A). All compounds in our library showed good free energy of binding to the target. However, to limit the docking results and to use a standard for other simulations, we selected only the first three best compounds on the basis of their binding free energy. From the docking results, COR480 (−8.8 Kcal/mol), COR482 (−8.8 Kcal/mol) and COR483 (−8.3 Kcal/mol) were the best three compounds for the S-glycoprotein trimerization region, while COR480 (−8.7 Kcal/mol), COR482 (−8.6 Kcal/mol) and COR1393 (−8.5 Kcal/mol) were the best compounds for the S-glycoprotein RBD. Finally, COR267 (−9.4 Kcal/mol), COR437 (−8.8 Kcal/mol) and COR1461 (−8.8 Kcal/mol) showed the best free energy of binding for the 3CLpro target.

To search for new compounds that could potentially block viral trimerization, we performed a virtual screening in our library within the S-glycoprotein region involved in trimer formation. Docking simulations and interaction network analyses provided us with useful insights. COR480, COR482 and COR483 exhibited good binding free energy at the target, from −9.4 to −8.8 Kcal/mol, and triggered both polar bonds and hydrophobic interactions with key residues for the formation of S-glycoprotein trimers [33]. cMD simulations and structural and energy analyses showed good stability for our ligands in the binding pocket of the target; moreover, each ligand exhibited a notable interaction energy along the trajectory. Our results showed that COR480, COR482 and COR483 spontaneously bind to the target with high affinity and that they may be able to disrupt the assembly of the quaternary structure of the SARS-CoV-2 S-glycoprotein, thereby affecting viral infection.

To understand whether the compounds in our collection are able to disrupt and/or prevent the interaction between S-glycoprotein and hACE2, we performed a molecular docking simulation within the SARS-CoV-2 RBD binding pocket that interacts with hACE2. COR480, COR482 and COR1393 were selected as the top three compounds because they had the lowest binding free energy for the target, ranging from −8.7 to −8.5 Kcal/mol. Interestingly, they formed nonpolar interactions, halogen bonds and hydrogen bonds within the target binding pocket. In particular, COR480 formed a hydrophobic interaction with Tyr-453 and a halogen bond with Gln-409, COR 482 formed two hydrogen bonds with Tyr-453 and Arg-403, and COR 1343 formed a hydrophobic interaction with Tyr-453 and a hydrogen bond with Arg-403. Previous work has identified Gln-409, Tyr-453 and Arg-403 as key residues in the receptor binding motif (RBM), both for binding to hACE2 and for binding to SARS-CoV-2 spike protein inhibitors [34]. To assess the absence of artefacts in our analysis, the structural integrity of the protein backbone was calculated during the simulations. We observed limited differences between the RMSD of the spike trimer, spike RBD and 3CLPro both in the free state and in complex with their ligands, ruling out the presence of irrational structural changes of the backbone. Interestingly, the spike trimer/COR 480 complex (Appendix A) and the spike RBD/COR 482 and 1393 complexes (Appendix A) showed distinct fluctuations in the MD simulations compared to the RMSD trend of their free state. This suggests that the compounds are able to alter the conformational stability of the protein, which likely affects the biological function of the target and its binding to hACE2. We noted apparent fluctuations in the RMSD trend of the binding poses of the spike trimer/COR 480 (Appendix A) and spike RBD/COR 482 (Appendix A) complexes. Such fluctuations can be explained by the different chemical and physical properties of the compounds and by the different docking–binding positions on the target. To confirm the ability of our compounds to bind to their target, a Coul-SR and LJ-SR energy analysis was performed. We observed that all ligands spontaneously bind to the target with a good energy. Overall, this suggests that our compounds prevent the recognition of hACE2 and, thus, bind to the SARS-CoV-2 spike protein and prevent the virus from entering the body.

To identify potential 3CLpro inhibitors in our library, we performed rational docking at the 3CLpro binding site of a known inhibitor (baicalein). The results of the virtual screening showed us that the top three compounds were COR267, COR437 and COR1461, with an energy score of −9.4 to −8.8 Kcal/mol. Interestingly, our compounds exhibited a broader interaction network than baicalein (Appendix A), forming hydrogen bonds and hydrophobic interactions with both the catalytic dyad residues and the 3CLpro residues that bind baicalein in the complex. To obtain further evidence for the potential inhibitory effect of our compounds on the target, we compared the docking results of COR267, COR437 and COR1461 with lopinavir and ritonavir (two experimental inhibitors of 3CLpro). After analyzing the docking results and interaction network results, we found that our compounds bound in the binding pocket of the target with a lower free energy of binding (from −9.4 to −8.8 Kcal/mol) than lopinavir and ritonavir (from −8 to −6.4 Kcal/mol), with several interactions triggered with the same binding residues of the two compounds, suggesting a possibly similar mechanism of action on the target. To rule out the presence of artefacts in our simulation, the RMSD values of the protein backbone and binding sites were evaluated. The 3CLPro/COR 267 and 437 complexes had RMSD values for the backbone that were higher than those in their free state (Appendix A) and comparable to the 3CLPro/ritonavir and lopinavir complexes (Appendix A), indicating a similar effect and mechanism of action on the target. The RMSD analyses obtained from the cMD simulations showed the good structural stability of the poses of our compounds docked to the target. The cMD simulation studies confirmed our evidence; indeed, from structural and energy analyses, we observed both good stability of the binding poses and very favorable interaction energy of the ligands with the target. Surprisingly, our compounds showed structural stability and interaction energy comparable to lopinavir and ritonavir. These results could provide an important basis for proposing our compounds as lead compounds for the synthesis of chemical analogues, such as a SARS-CoV-2 3CLpro inhibitor. Additional in silico and in vitro analyses should be performed to investigate further aspects, such as toxicity and the ability of our analogues to bind selectively to their target. On the basis of the computational results obtained and demonstrated in this study, we have selected the compounds COR 437, COR 480 and COR 482 as scaffold candidates for future in vitro analyses, and we believe that they are most likely to be suitable for clinical applications. However, while docking and cMD simulations are reliable computational methods to predict and propose potentially active compounds that can be used against a biological target, and to help understand their potential mechanism of action, and despite the enormous progress in this field of research, experimental tests are necessary to verify the in silico results. By considering this computational study, research groups may take advantage our findings in order to plan new in silico and in vitro procedures for the identification of novel potential lead compound classes to use as multitarget therapeutic strategy for combating COVID-19.

## 5. Conclusions

In this work, we presented a SARS-CoV-2 multi-target-directed ligand approach as a therapeutic strategy for COVID-19 disease. We focused our attention on two different targets of the virus: S-glycoprotein and 3CLpro. Our study was based on three different strategies to combat the infection by SARS-CoV-2: (i) blocking the trimerization of viral S protein, (ii) preventing the interaction of SARS-CoV-2 S protein with hACE2 and (iii) inhibiting the biological function of 3CLPro. To perform our study, a total of 1764 compounds that we synthesized ourselves were subjected to virtual screening with molecular docking against SARS-CoV-2. The results of the docking simulations showed that several compounds were able to spontaneously bind to their targets with high affinity. However, in order to apply a standard criterion for each target, only the first three best compounds (based on their free energy of binding) were selected for each receptor. The best compounds were identified as COR480, COR482 and COR 483 for the first strategy; COR480, COR482 and COR1393 for the second strategy; and COR267, COR437 and COR1461 for the third strategy. The docking simulations showed that our compounds were able to trigger a broad interaction network within their target binding pocket. Interestingly, as described in Section 4, some compounds were able to form hydrophobic interactions, halogen bonds and hydrogen bonds with crucial residues of the targets, suggesting further potential inhibitory activity against the receptors. To confirm the docking evidence, 100 ns MD simulations were performed for all atoms. All compounds showed favorable overall interaction energy and stable binding poses in their binding pockets, confirming the reliability of the initial docking poses. The spike trimer/COR 480 complex, the spike RBD/COR 482 and 1393 complexes, and the 3CLPro/COR 267 and 437 complexes exhibited strong backbone RMSD fluctuations compared to their free state during MD simulation, indicating a potential perturbation effect on the structural stability of the protein. Interestingly, the 3CLPro/COR 267 and 437 complexes exhibited backbone RMSD values comparable to the 3CLPro/ritonavir and lopinavir complexes, suggesting a similar effect and mechanism of action on the target.

On the basis of the evidence provided, the compounds COR 437, COR 480 COR 482 are recommended as scaffolds for the synthesis of chemical analogues to be tested in vitro for proposed activity against COVID-19. Our results may provide the important insights into SARS required to make spike protein and 3CLPro targeting drugs to facilitate future design and synthesis of new candidates against COVID-19.

## Figures and Tables

**Figure 1 biology-11-00465-f001:**
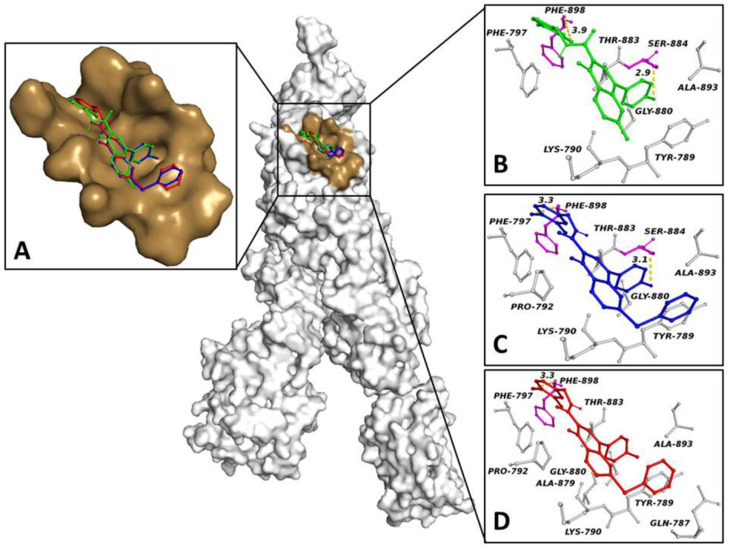
S-Glycoprotein trimerization region and compound binding sites. Surface representation of the S protein in complex with ligands. (**A**) The pocket surface patch is depicted in brown, while the ligands are shown in colored balls and sticks. (**B**–**D**) Structural representations of the (**B**) S-glycoprotein-COR 480, (**C**) S-glycoprotein-COR 482 and (**D**) S-glycoprotein-COR 483 complexes resulting from the docking simulations. Residues forming direct interactions with the ligands are shown as gray (hydrophobic interaction) and magenta (halogen bond) balls and sticks. Halogen bonds are indicated with a yellow dashed line. The number close to the dashed lines represents the bond length.

**Figure 2 biology-11-00465-f002:**
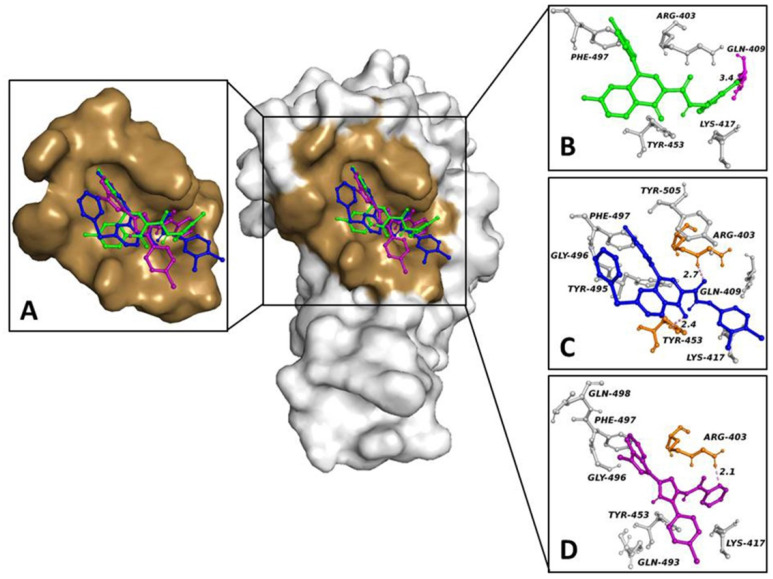
RBD region and compound binding sites. Surface representation of S-glycoprotein RBD in complex with ligands. (**A**) The pocket surface patch is depicted in brown, while the ligands are shown in colored balls and sticks. (**B**–**D**) Structural representations of the (**B**) RBD-COR 480, (**C**) RBD-COR 482 and (**D**) RBD-COR 1393 complexes resulting from the docking simulations. Residues forming direct interactions with the ligands are shown as gray (hydrophobic interaction), magenta (halogen bond) and orange (hydrogen bond) balls and sticks. Halogen and hydrogen bonds are indicated with yellow and pink dashed lines, respectively. The number close to the dashed lines represents the bond length.

**Figure 3 biology-11-00465-f003:**
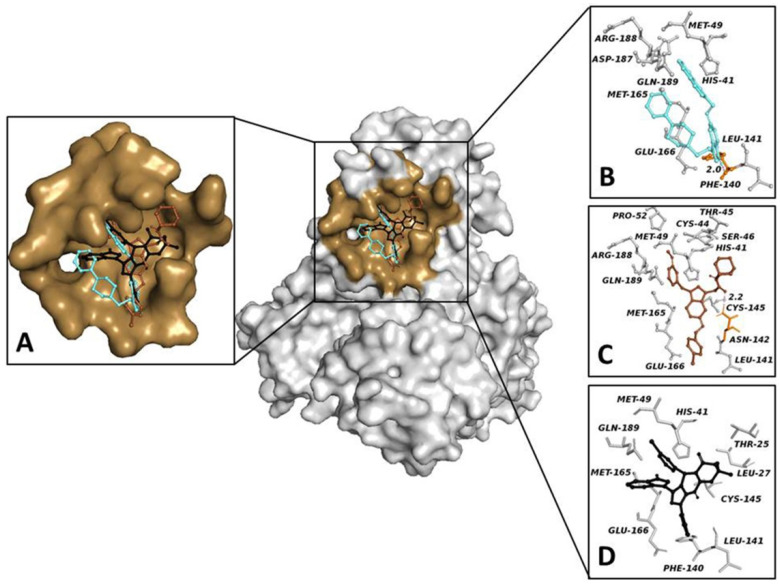
This figure shows 3CLpro and compound binding sites. Surface representation of the structure of the 3CLpro in complex with ligands. (**A**) The pocket surface patch is depicted in brown, while the ligands are shown in colored balls and sticks. (**B**–**D**) Structural representations of the (**B**) 3CLpro binding pocket-COR 267, (**C**) 3CLpro binding pocket-COR 437 and (**D**) 3CLpro binding pocket-COR 1461 complexes resulting from docking simulations. Residues forming direct interactions with the ligands are shown as gray (hydrophobic interaction) and orange (hydrogen bond) balls and sticks. Hydrogen bonds are indicated with pink dashed lines. The number close to the dashed lines represents the bond length.

## Data Availability

Not applicable.

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
