# Peer review of "In Silico Multi-Target Approach Revealed Potential Lead Compounds as Scaffold for the Synthesis of Chemical Analogues Targeting SARS-CoV-2"

_biology, 2022, doi:10.3390/biology11030465_

Round 1
Reviewer 1 Report
This is an interesting paper, and the authors have performed in silico analysis of a chemical library of around 1700 molecules to propose potential protein inhibitors of COVID-19. The paper is generally well written and structured.
Prior to the publication of this manuscript, minor observations should be attended:
- Lines 21-22: “The absence of drug-like molecules gave increase to an extensive multiplicity of strategies employed to fight this pandemic.” In my opinion, it would be great if the authors can better motivate the study.
- Lines 83-84: I would suggest adding a citation to this text "Nevertheless, the mortality remains high, showing that treatment with an antiviral drug 83 alone is not likely to be enough".
- Lines 361-363: “These findings could represent significant basis for proposing our compounds as lead compounds as scaffold for the synthesis of chemical analogues like-inhibitor targeting SARS-CoV-2 3CLpro” At the end of this sentence, I would suggest convincing the reader how these findings are helpful.
- I would suggest improving the quality of supplementary figures to increase the readability, and the figures in the original manuscript if possible.
- Line 158: Please add version number for LigPlot++.
- Please check the tools used in this study to leave space after the tool name. For instance, change PyMOD2.0 to PyMOD 2.0.
- Change "3.3. 3CLpro: Virtual screening and cMD" to "3.3. 3 CLpro: Virtual screening and cMD"
Author Response
All authors thank the Reviewer for his helpful advices.
This is an interesting paper, and the authors have performed in silico analysis of a chemical library of around 1700 molecules to propose potential protein inhibitors of COVID-19. The paper is generally well written and structured.
Prior to the publication of this manuscript, minor observations should be attended:
Point 1. Lines 21-22: “The absence of drug-like molecules gave increase to an extensive multiplicity of strategies employed to fight this pandemic.” In my opinion, it would be great if the authors can better motivate the study.
Response 1. Huge efforts have been done to overcome the COVID-19 pandemic as well as different strategies have been proposed and adopted. In this scenario, would be very useful the identification of novel approaches to use against the infection by SARS-CoV-2. As suggested by Reviewer, we rewrote the sentence (Lines 21-23) in order to motivate the study.
Point 2. Lines 83-84: I would suggest adding a citation to this text "Nevertheless, the mortality remains high, showing that treatment with an antiviral drug 83 alone is not likely to be enough".
Response 2. Unfortunately, drug alone-based treatments proposed to fight COVID-19 showed a limited or poor efficacy. Approaches multitarget-directed-ligand may be a good strategy to adopt for COVID-19. We are agree with Reviewer, and we added some references to the text in lines 83-84.
Point 3. Lines 361-363: “These findings could represent significant basis for proposing our compounds as lead compounds as scaffold for the synthesis of chemical analogues like-inhibitor targeting SARS-CoV-2 3CLpro” At the end of this sentence, I would suggest convincing the reader how these findings are helpful.
Response 3. To propose new strategies and scaffold for the synthesis of chemical analogues like-inhibitor targeting SARS-CoV-2 directed against different target of the virus, may be very useful to look for therapies to overcome this pandemic. Here, we propose, for the first time, a novel scaffold class showing a potential inhibitory activity against SARS-CoV-2 targets. As suggested by Reviewer, we added a sentence (lines 363-366) to highlight the importance of our findings. Furthermore, the authors added the conclusion section to further support the evidences observed in this study.
Point 4. I would suggest improving the quality of supplementary figures to increase the readability, and the figures in the original manuscript if possible.
Response 4. To provide figures with high resolution is a very important point to increase and improve the readability of a paper. The authors improved the quality of all figures, both in supplementary materials and in original manuscript. Now, all figures of supplementary materials, from a starting resolution of 96 dpi have been modified with a resolution of 600 dpi (with maximum quality of the image). Similarity, all figures in the original manuscript, from a starting resolution of 300 dpi have been modified with a resolution of 600 dpi (with maximum quality of the image).
Point 5. Line 158: Please add version number for LigPlot++.
Response 5. The version number for LigPlot++ was added to the line 163.
Point 6. Please check the tools used in this study to leave space after the tool name. For instance, change PyMOD2.0 to PyMOD 2.0.
Response 6. We checked and added a space after the tool name used in this study (line 128).
Point 7. Change "3.3. 3CLpro: Virtual screening and cMD" to "3.3. 3 CLpro: Virtual screening and cMD"
Response 7. The statement "3.3. 3CLpro: Virtual screening and cMD" was changed to "3.3. 3 CLpro: Virtual screening and cMD" (line 266).

Reviewer 2 Report
Developing better drugs that can treat the SARS-COVID-19 infection is of utmost priority to the entire world because of the lack of small molecule inhibitors that can treat the patients with severe symptoms leading to death and the rapid variants of SARS-COVID that are emerging with time. The authors have used an in silicoapproach to identify potential lead molecules that can
be tested for antiviral activity using homology modeling, docking and cMD simulations. Using a predesigned library of compounds with drug-like small molecule analogs, they have reported 3 top hits for SARS-CoV-2 S-glycoprotein trimerization region, SARS-CoV-2 RBD/hACE2 interaction region and the inhibitor binding region of 3C-like protease (3CLpro).
Strengths:
- This virtual screening study is well designed and executed.
- The methods used are well explained
- The results are described well and validated in comparison to the existing inhibitors wherever there is a known inhibitor
Weaknesses:
- In discussion, the authors propose to synthesize more analogs of the lead molecules they have suggested. But the first goal should be test these hits in functional assays to show that they have the antiviral activity using bioanalytical assays such as inhibition of cell fusion used in the reference (doi: 1002/jmv.25985) or the assays used to validate Lopinavir and Ritonavir as antiviral agents.
- The authors should also discuss if any structural analogs of the lead molecules existing in the library but did not have comparable binding free energy
- The English sentence structure requires professional scientific editing. For example: “The absence of drug-like molecules gave increase to an extensive multiplicity of strategies employed to fight this pandemic. “
Author Response
All authors thank the Reviewer for his helpful advices.
Developing better drugs that can treat the SARS-COVID-19 infection is of utmost priority to the entire world because of the lack of small molecule inhibitors that can treat the patients with severe symptoms leading to death and the rapid variants of SARS-COVID that are emerging with time. The authors have used an in silicoapproach to identify potential lead molecules that can
be tested for antiviral activity using homology modeling, docking and cMD simulations. Using a predesigned library of compounds with drug-like small molecule analogs, they have reported 3 top hits for SARS-CoV-2 S-glycoprotein trimerization region, SARS-CoV-2 RBD/hACE2 interaction region and the inhibitor binding region of 3C-like protease (3CLpro).
Point 1. In discussion, the authors propose to synthesize more analogs of the lead molecules they have suggested. But the first goal should be test these hits in functional assays to show that they have the antiviral activity using bioanalytical assays such as inhibition of cell fusion used in the reference (doi: 1002/jmv.25985) or the assays used to validate Lopinavir and Ritonavir as antiviral agents.
Response 1. Docking and Molecular dynamics simulation are good computational methods to predict and to propose potential active compounds for a target as well as their potential mechanism of action. Despite the enormous progress made in this research field, the biological verification is needful to verify in silico results. The aim of this study, was not the identification of experimental inhibitors of SARS-CoV-2 targets, but, to propose potential lead compounds as scaffold for the synthesis of chemical analogues targeting SARS-CoV-2, providing new findings for the drug discovery in order to plan new in silico and in vitro procedures for the identification of novel lead compound classes to use as multitarget therapeutic strategy for the COVID-19 disease. However, we completely are agree with the Reviewer comment, and the biological evaluation of compounds will be performed in the near future in order to confirm the computational results from an experimental point of view. The authors added a new statement to clarify the aim of this study (line 363-366).
Point 2. The authors should also discuss if any structural analogs of the lead molecules existing in the library but did not have comparable binding free energy.
Response 2. Docking simulation is a computational method that allow us to predict the binding energies between biological systems. In this study we screened >1700 compounds on three different SARS-CoV-2 targets in order to identify the best compounds for each target. The virtual screening, is a good approach to use to identify the best binding poses of compounds belonged to a wide library. Nevertheless, the docking simulation has limitations, from them: i) the whole protein surface during the docking simulation is rigid ii) the solvent is absent iii) the algorithm scoring functions. Recently many efforts have been made to deal with the flexibility of the receptor [1], however, flexible receptor docking, especially backbone flexibility in receptors, as well as, a correlation of different scoring functions to obtain a score more reliable, still presents a major challenge for available docking methods. In our study, to consider further features of our best compounds on the target, we performed classical molecular dynamics (cMD). From cMD results, we noted that compounds showed any differences between the docking and cMD binding energy. Such differences could be caused from limitations of the docking simulation. In fact, in cMD we consider the dynamicity of biological systems, the presence of the solvent and the target/ligand binding score function is based on their interaction energy. However, the aim of this study was to identify and provide new insights of potential lead compounds as scaffold for the synthesis of chemical analogues targeting SARS-CoV-2, comparing their activity and mechanism of action with known experimental inhibitors of SARS-CoV-2. We completely are agree with Reviewer about the discussion if any structural analogues of the lead molecules existing in the library but did not have comparable binding free energy. To further give strength to our simulations, we preferred to discuss the activity of our compounds compared to known SARS-CoV-2 experimental inhibitors to avoid excessive computational speculations. However, as proposed by Reviewer, following to future experimental studies, we’ll carry out new experiments comparing our best compounds with any structural analogues of the lead molecules existing in the library but did not have comparable binding free energy in order to confirm or provide new findings about our work.
Point 3. The English sentence structure requires professional scientific editing. For example: “The absence of drug-like molecules gave increase to an extensive multiplicity of strategies employed to fight this pandemic. “
Response 3. All authors reviewed and modified several sentences of the paper, for instance: “The absence of drug-like molecules gave increase to an extensive multiplicity of strategies employed to fight this pandemic.” to “Identifying a multi-target-directed ligand approach would open up new opportunities for drug discovery to combat COVID -19 (lines 22-23).”. Furthermore, the article has been revised by a professional scientific editing.
References:
- Meng, X.-Y.; Zhang, H.-X.; Mezei, M.; Cui, M. Molecular Docking: A Powerful Approach for Structure-Based Drug Discovery. Curr. Comput. Aided-Drug Des. 2012, doi:10.2174/157340911795677602.

Reviewer 3 Report
- There are typing errors, small vs capital letters errors, punctuation errors throughout the manuscript.
- The World Health Organization 37 (WHO) declared COVID-19 could be characterized as a pandemic on the 12th of March 38 2020 - this sentence may bring confusion. COVID19 is a pandemic.
- There are some unsuitable terms being used throughout the manuscript. It is suggested to send the manuscript to a professional proof reader. (e.g: ....but some types of patients, especially the old ones...) More suitable terms should be used instead.
- This sentence is confusing (..Presently, there are several encouraging therapeutic candidates, that are being evaluated against COVID-19 with emergency-use authorization by the FDA and NIH such as 74 remdesivir....) - Remdesivir is approved or is currently being evaluated?
- ..."The energy of the system was minimized as previously decribed" please add the reference.
- Adding a figure on the flow of the study will help in understanding.
- The method section is rather confusing. I would like to suggest to separate the in silico method section according to the technique - e.g 2.2.1 Molecular docking, 2.2.2 Molecular dynamics, etc.
- Figure 1 caption described the color code for each interaction (hydrogen bonding, halogen bonding and hydrophobic interaction). Based from the discussion, there is non hydrogen bonds formed. It is suggested to remove the description of hydrogen bonding from the caption in Figure 1.
- Based on the information in Table S2 (supplementary file), the binding energy of COR483 and COR925 is exactly the same. Why did the authors chose COR483 and not COR925? Same questions for the other two receptors as presented in Table S3 and S4.
- The image in Figure 2B is not aligned to the discussion "COR480 was involved in a halogen bond between the oxygen atom 230 (O1) of the carbonyl group and the hydroxyl of Tyr-453 side chain, and the oxygen atom 231 (O2) of the other carbonyl group and the Arg-403". Please recheck.
- Figure 2C showed hydrogen bonding but not described in the text. Hydrogen bonding is important, thus should be described.
- Figure 2D - Asn 501 is mentioned in the text but not present in the figure, whilst Gln 498 is available in the figure but not mentioned in the text. This is confusing, suggest to recheck.
- "....From docking and interaction network results we noted that our compounds bound within target binding pocket with a binding free energy higher than Lopinavir and Ritonavir .." - higher or lower binding energy?
- "....we observed both a good stability of the binding poses and a high interaction energy of ligands with the target." - is high interaction energy favourable? should it be high affinity instead? or low energy?
- The discussion on the cMD can be improved. There are fluctuations in the RMSD of the complex, and some complexes showed obvious fluctuations throughout the MD simulation. Can the authors explain the relation of the fluctuations with the stability of the complexes? (Fig S2)
- The authors did not offer any conclusion in the manuscript.
Author Response
All authors thank the Reviewer for his helpful advices.
There are typing errors, small vs capital letters errors, punctuation errors throughout the manuscript.
Point 1: The World Health Organization 37 (WHO) declared COVID-19 could be characterized as a pandemic on the 12th of March 38 2020 - this sentence may bring confusion. COVID19 is a pandemic.
Response 1: The authors are agree with the Reviewer about the confusion that may bring the sentence: “The World Health Organization 37 (WHO) declared COVID-19 could be characterized as a pandemic on the 12th of March 38 2020”. The sentence was rewrote as: “In March 2020, the World Health Organisation (WHO) declared COVID -19 pandemic (lines 21-22).”.
Point 2: There are some unsuitable terms being used throughout the manuscript. It is suggested to send the manuscript to a professional proof reader. (e.g: ....but some types of patients, especially the old ones...) More suitable terms should be used instead.
Response 2: All authors reviewed and modified several sentences of the paper. For instance, the sentence: “but some types of patients, especially the old ones” was replaced as: “The disease ends in a few days, but some types of patients, especially the elderlies and people with co-existing illness…” reported in the line 53, as suggested by the Reviewer. However, the manuscript has been revised by professional scientific editing.
Point 3: This sentence is confusing (..Presently, there are several encouraging therapeutic candidates, that are being evaluated against COVID-19 with emergency-use authorization by the FDA and NIH such as 74 remdesivir....) - Remdesivir is approved or is currently being evaluated?
Response 3: The authors presented unclearly their statement about remdesivir's status as COVID-19 treatment. As proposed by the Reviewer, we rewrote the sentence from: “Presently, there are several encouraging therapeutic candidates, that are being evaluated against COVID-19 with emergency-use authorization by the FDA and NIH such as remdesivir” to “Presently, there are several encouraging therapeutic candidates, that are being evaluated against COVID-19 with emergency-use authorization by the FDA and NIH such as remdesivir, a FDA-approved inhibitor [10] of the viral RNA synthesis…(lines 73-76)”.
Point 4: ..."The energy of the system was minimized as previously decribed" please add the reference.
Response 4: The authors presented unclearly their statement: “The energy of the system was minimized as previously described", in this sentence, the authors wanted to report the same protocol of energy minimization described in the lines 132-141 in this manuscript. The sentence was rewrote as: “The energy of the system was minimized as above-mentioned” (line 141).
Point 5: The method section is rather confusing. I would like to suggest to separate the in silico method section according to the technique - e.g 2.2.1 Molecular docking, 2.2.2 Molecular dynamics, etc.
Response 5: To clearly explain the procedures adopted in a scientific article is a very important point to increase the readability of the paper. The authors are agree with the Reviewer to separate in silico methods according to the technique, in order to provide clarity about the methods applied in this work. The authors added the subparagraphs “in silico methods” section, as suggested by the Reviewer.
Point 6: Figure 1 caption described the color code for each interaction (hydrogen bonding, halogen bonding and hydrophobic interaction). Based from the discussion, there is non hydrogen bonds formed. It is suggested to remove the description of hydrogen bonding from the caption in Figure 1.
Response 6: To describe a clear caption of a figure is essential to explain its representation. Indeed, as highlighted by the Reviewer, no hydrogen bond was present in Figure 1, this may confuse the reader. The authors removed the description of hydrogen bonding from the caption in Figure 1.
Point 7: Based on the information in Table S2 (supplementary file), the binding energy of COR483 and COR925 is exactly the same. Why did the authors chose COR483 and not COR925? Same questions for the other two receptors as presented in Table S3 and S4.
Response 7: Docking simulation is a computational method able to simulate the interaction between biological systems. Different docking software use different algorithms to dock a ligand and to score its affinity on the target. In this study, we performed a virtual screening in order to obtain hit compounds belonged to a wide library. Hit compounds showed a very little difference between them in score terms. However, to restrict docking results and to adopt and use a standard criterion for other simulations, we selected only the first three best compounds, on the basis of their binding free energy on the target. To confirm docking results and to show the reliability of compound selection, we performed classical molecular dynamics (cMD) considering structural and energetic features of compounds on their target. From cMD results, we observed that our compounds bound with high affinity and good stability within target binding pocket, confirming docking results and the reliability of our hit compound selection. This standard criterion was used for other two receptors as presented in Table S3 and S4. The authors clarified this point in the discussion (line 318) and conclusion (line 414) sections.
Point 8: The image in Figure 2B is not aligned to the discussion "COR480 was involved in a halogen bond between the oxygen atom 230 (O1) of the carbonyl group and the hydroxyl of Tyr-453 side chain, and the oxygen atom 231 (O2) of the other carbonyl group and the Arg-403". Please recheck.
Response 8: Indeed, The image in Figure 2B is not aligned to the discussion "COR480 was involved in a halogen bond between the oxygen atom 230 (O1) of the carbonyl group and the hydroxyl of Tyr-453 side chain, and the oxygen atom 231 (O2) of the other carbonyl group and the Arg-403". The authors explained unclearly the sentence. As suggested by the Reviewer, the sentence was rechecked and rewrote (line 344).
Point 9: Figure 2C showed hydrogen bonding but not described in the text. Hydrogen bonding is important, thus should be described.
Response 9: The hydrogen bond is an important interaction between two biological systems, in our case, between protein and small molecule, which could be key for the mechanism of action of the compound and the inhibition of the biological function of the protein. To describe its effect is crucial to explain the effect of a small molecule against the target. The authors explained unclearly the sentence. As suggested by the Reviewer, the sentence was rechecked and rewrote (lines 344-348).
Point 10: Figure 2D - Asn 501 is mentioned in the text but not present in the figure, whilst Gln 498 is available in the figure but not mentioned in the text. This is confusing, suggest to recheck.
Response 10: In figure 2D, the authors mentioned in the text Asn-501 but not present in the figure, while, Gln-498 is available in the figure but not mentioned in the text. The mistake was corrected (line 243).
Point 11: "....From docking and interaction network results we noted that our compounds bound within target binding pocket with a binding free energy higher than Lopinavir and Ritonavir .." - higher or lower binding energy?
Response 11: The authors explained unclearly the sentence. As suggested by the Reviewer, the sentence was rechecked and rewrote (lines 376-377).
Point 12: "....we observed both a good stability of the binding poses and a high interaction energy of ligands with the target." - is high interaction energy favourable? should it be high affinity instead? or low energy?
Response 12: The authors explained unclearly the sentence. As suggested by the Reviewer, the sentence was rechecked and rewrote (lines 387-388).
Point 13: The discussion on the cMD can be improved. There are fluctuations in the RMSD of the complex, and some complexes showed obvious fluctuations throughout the MD simulation. Can the authors explain the relation of the fluctuations with the stability of the complexes? (Fig S2)
Response 13: To clearly explain the results obtained in a scientific work is a crucial point to increase the readability of the paper. The authors are agree with the Reviewer and provided the relation of the fluctuations with the stability of the complexes as reported in Figure S2. Furthermore, to further improve the discussion on the cMD, a comparison between 3CLPro/COR 267 and 437 complexes (Figure S2) and 3CLPro/Lopinavir and Ritonavir complexes (Figure S3) was reported in discussion section (lines 350-361; 381-389).
Point 14: The authors did not offer any conclusion in the manuscript.
Response 14: The conclusion section is a very important section because allow to readers to have a complete overview of the study as well as of results achieved. As suggested by the Reviewer, the authors added the conclusion section in the manuscript (lines 404-436).
Point 15: Adding a figure on the flow of the study will help in understanding.
Response 15: To provide a schematic workflow used in this study, to perform the different bioinformatics methods, is very useful to help in understanding each step of the study. The authors added the flow of the study in the Figure S2.

Reviewer 4 Report
In the current scenario of the pandemic, the findings are timely and promising. Computational tools play a key role in drug development. The authors identified compounds targeting SARS-CoV-2 3CLpro as new template for antiviral drugs. Some points need to be clarified.
- Some statements are contradictory. Review lines 61-62 and 71-72.
- The name and structure of the 1764 compounds from the library used in the virtual screening should be included in the supplementary material. This information is valuable.
- Line 54: Change Countries to countries
- The in silico findings, although promising, must be posteriorly validated experimentally (in the future). Perspectives and limitations of this study should be included in the discussion sections. Which compounds do the authors consider most likely to be clinically translated?
- Toxicity and selectivity is a challenge in drug development. Did the authors assess the ability of these compounds to attach to other non-target molecules? The authors are invited to discuss this.
Author Response
All authors thank the Reviewer for his helpful advices.
In the current scenario of the pandemic, the findings are timely and promising. Computational tools play a key role in drug development. The authors identified compounds targeting SARS-CoV-2 3CLpro as new template for antiviral drugs. Some points need to be clarified.
Point 1. Some statements are contradictory. Review lines 61-62 and 71-72
Response 1. The authors wanted highlighting the progresses obtained in the drug repurposing field aimed to identify antiviral drugs for COVID-19 (lines 61-62) and their effect to fully overcome COVID-19 (lines 71-72). Indeed, the authors are completely agree with the Reviewer that the statements are contradictory. To clarify our concept, we moved the sentence from lines 61-62 to the line 76. The sentence to lines 71-72 was rewrote in lines 71-73.
Point 2. The name and structure of the 1764 compounds from the library used in the virtual screening should be included in the supplementary material. This information is valuable.
Response 2. Several computational and experimental methods are based on the knowledge of 3D structures of compounds having activity against a biological target. In this scenario, to know the 3D structures of compounds is a crucial information to further perform studies and analyses. Unfortunately, the compound library used in this study, it contains several compounds undergo to patents or used for other projects that not allow their release. All authors are fully agree with the reviewer and are aware of the importance of this information, this is the reason why, that we provided the name and the 3D structures of the best 10 compounds of our study, to allow to other research groups to further investigate on this study and other studies.
Point 3. Line 54: Change Countries to countries
Response 3. The word was changed as required by Reviewer (line 55).
Point 4. The in silico findings, although promising, must be posteriorly validated experimentally (in the future). Perspectives and limitations of this study should be included in the discussion sections. Which compounds do the authors consider most likely to be clinically translated?
Response 4. Docking and Molecular dynamics simulations are good computational methods to predict and propose potential active compounds for a target as well as their potential mechanism of action. Despite the enormous progress made in this research field, the biological verification is needful to verify in silico results. The aim of this study, was not the identification of experimental inhibitors of SARS-CoV-2 targets, but, to propose potential lead compounds as scaffold for the synthesis of chemical analogues targeting SARS-CoV-2, providing new findings for the drug discovery in order to plan new in silico and in vitro procedures for the identification of novel lead compound classes to use as multitarget therapeutic strategy for the COVID-19 disease. However, we completely are agree with the Reviewer comment, and the biological evaluation of compounds will be performed in the near future in order to confirm the computational results from an experimental point of view.
The authors added the perspectives/limitations of this study and the compounds that the authors consider most likely to be clinically translated, in the discussion section (lines 389-403).
Point 5. Toxicity and selectivity is a challenge in drug development. Did the authors assess the ability of these compounds to attach to other non-target molecules? The authors are invited to discuss this.
Response 5. Huge milestones have been achieved in drug development field about the toxicity and selectivity of compounds used against biological targets. Unfortunately, this side is still a challenge in drug development. The aim of this study was focused on the identification of potential lead compounds as scaffold for the synthesis of chemical analogues targeting SARS-CoV-2 using in silico multi-target approach. On the basis of this work, we’ll start, in near future, with the synthesis of analogues, evaluating other features, such as: toxicity and selectivity. We are agree with the Reviewer about the importance of this aspect and the authors discussed this in the discussion section (lines 392-393).
